# CARE: Contextual Affinity Exploration with Twin Concordance for Graph Out-Of-Distribution Detection

## Abstract

This paper studies the problem of graph out-of-distribution (OOD) detection, which aims to identify anomaly graphs out of a graph dataset. Prior efforts usually focus on the utilization of topological structures with unsupervised graph learning to foster typical pattern recognition, which overlooks the semantic structure preserved in contextual affinity neighborhoods. Towards this end, we propose a novel approach named Contextual Affinity Exploration with Twin Concordance (`CARE`) for graph OOD detection. The core of `CARE` is to explore and exploit the contextual affinity of the graph data samples for discriminative graph representations. In particular, our `CARE` first builds a contextual affinity graph to depict the semantic structure in the hidden space. More importantly, we introduce high-order affinity to enhance geometric understanding of the structure by utilizing a meta-graph neural network. To enhance representation discriminability with high robustness, we introduce twin concordance learning, which not only minimizes the difference of affinity distributions across different views, but also encourages the consistency between contextually affinitive neighbors. Finally, we introduce a compression strategy to expand the decision boundary for enhanced separation between in-distribution and out-of-distribution graphs. Extensive experimental results demonstrate the superiority of our `CARE` across ten real datasets in comparison to various baselines.

## 1 Introduction

Graphs serve as an indispensable tools for handling complex relational data across diverse fields, including molecular graph (Hao et al., 2020), biological network (Borgwardt et al., 2005) and citation network (Wang et al., 2022), etc. Building on the powerful representation capabilities of deep learning, graph neural networks (GNNs) have emerged as a leading paradigm for encoding complex graph data (Kipf & Welling, 2017; Veličković et al., 2018; Xu et al., 2018). By leveraging a neighborhood-based message-passing mechanism to iteratively refine node representations, followed by a readout function to aggregate these updates into a comprehensive graph-level representation, GNNs are well-equipped to capture intricate relationships within the data and excel in various applications (Wu et al., 2020).

However, a key challenge with GNNs lies in their reliance on the i.i.d. assumption, which posits that testing data is sampled from the same In-Distribution (ID) as the training set. In real-world scenarios, especially when labeled graph data is limited, this assumption often fails (Ju et al., 2024b). As a result, GNNs typically struggle with out-of-distribution (OOD) graph samples that were not seen during training. Ideally, a trustworthy graph machine learning system should not only learn from known ID samples, but also recognize unknown OOD inputs during the inference phase (Wu et al., 2022). This underscores the importance of graph OOD detection, which helps determine whether a graph is from the ID or OOD, allowing the model to take appropriate precautions.

Recently, there has been a variety of methods proposed for OOD detection, especially in images (Hendrycks & Gimpel, 2017; Sehwag et al., 2021) or text domain (Zhou et al., 2021). For graph-structured data, detecting OOD graph samples is inherently more difficult due to both the structural and property patterns that can vary across different domains. Several pioneering stud-

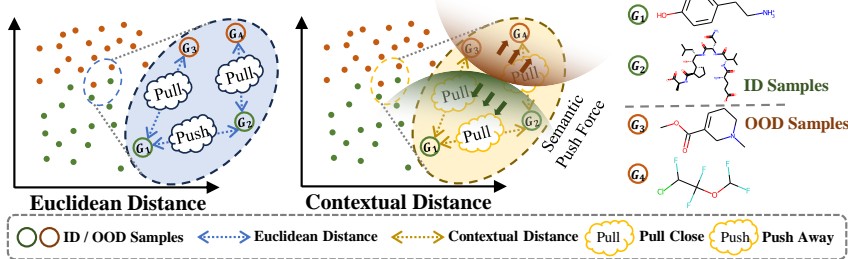

Figure 1: Comparison between the conventional Euclidean neighborhood and the contextually affinitive neighborhood. Under Euclidean distance metrics, cross-dataset samples with analogous feature patterns may exhibit proximity in the feature space. In contextually affinitive space, samples with diverse topological structures from the same dataset tend to show semantical similarities in a shared underlying distribution.

ies have begun to explore graph OOD detection. For example, GraphDE (Li et al., 2022) models a graph-generative process to capture distribution shift for graph OOD detection. GOOD-D (Liu et al., 2023), AAGOD (Guo et al., 2023) and GOODAT (Wang et al., 2024) further concentrate on the scarcity of graph OOD samples and propose unsupervised methods that are trained from scratch, post-hoc and at test-time respectively. HGOE (He et al., 2024) synthesizes outliers within ID graph samples to assist OOD detection training.

Despite the encouraging performance of these graph OOD detection methods, we argue they overlook the semantic structure preserved in contextual affinity neighborhoods. In fact, when two graph samples are drawn from the same data distribution, they are contextually affinitive, which means they may share similar relationships with a set of reference samples. As shown in Figure 1, $G_1$ and $G_2$ exhibit diverse topological structures (i.e., Benzene and Carbon Chain) results in their greater distant in Euclidean space. However, these samples belong to the same data distribution depicting contextual affinity and remain semantically distinct from samples in other datasets, despite they shows similarities in topological structures (i.e., $G_1$ and $G_3$, $G_2$ and $G_4$). Therefore, there is a pressing need for an approach that could leverage the semantic structure preserved within neighborhood contexts for more effective graph OOD detection.

Towards this end, we propose a novel approach termed **C**ontextual **A**ffinity Explo**R**ation with Twin Concordanc**E** (`CARE`) for graph OOD detection, which employs an efficient retrieval process to mine more informative neighbors and encourages neighborhood concordance to learn a discriminative ID sample boundary. Specifically, given the input graph, we define the reference graphs as the current batch and estimate the corresponding high-order affinity to excavate more semantically relevant neighbors in a contextually affinitive space. Then, we consider the graph and its augmentation as two views, and group the graph into semantically meaningful clusters. Based on this, neighborhood-based concordance learning is proposed to explore the preserved semantic structure, enabling the model to produce a consistent and discriminative boundary for different underlying semantics within ID graphs. Finally, to achieve a more refined separation between ID and OOD graph samples, we introduce a decision boundary compression strategy for ID graphs, and thus, any test graph sample located out of the semantic boundary can be detected as an OOD graph.

In a nutshell, the main contributions of this paper can be summarized as follows: ❶ *New Perspective.* We propose a new perspective of leveraging contextual affinity to enhance graph OOD detection, which provides a clear understanding of the semantic structure in the hidden space. ❷ *Novel Methodology.* Our `CARE` not only enhances the geometric understanding with contextual high-order affinity, but also leverages twin affinity-aware concordance learning for robust and discriminative graph representations. ❸ *State-of-the-art Performance.* We conduct extensive experiments on a variety of benchmark datasets to evaluate our `CARE`.

## 2 RELATED WORK

**Graph Neural Networks.** Graph neural networks (GNNs) offer a promising approach for handling graph-structured data and are gaining increasing attention in recent times (Hamilton et al., 2017; Veličković et al., 2018; Xu et al., 2018). From the perspective of learning paradigm, GNNs can be

broadly categorized into two types: 1) Spectral-based GNNs and 2) Spatial-based GNNs. Spectral-based GNNs aim to learn graph convolution through spectral graph theory (Kipf & Welling, 2017; Wu et al., 2019). For example, GCN (Kipf & Welling, 2017) performs convolution using the graph Laplacian matrix to define filters that capture the graph structure. Spatial-based GNNs concentrate on aggregating and transforming local information (Veličković et al., 2018; Xu et al., 2018). For example, GIN (Xu et al., 2018) learns node representations by aggregating features from neighboring nodes in a manner that is sensitive to the graph structure, facilitating effective discrimination between different topologies. Despite the growing popularity of GNNs, several concerns have recently emerged. In this paper, we focus on OOD detection, aiming to provide a reliable learning framework for identifying OOD samples.

**Out-of-distribution Detection.** Out-of-distribution (OOD) detection involves identifying samples that significantly differ from the training distribution, helping to prevent model misclassifications and enhance robustness. Despite their success in image (Liang et al., 2018; Hendrycks & Gimpel, 2022) and text (Zhou et al., 2021) data, OOD detection in graph data still faces considerable challenges due to the complex topological structures that hinder the effect pattern recognition. AAGOD (Guo et al., 2023) proposed a parameterized amplifier to highlight prominent patterns, and thereby enhancing the distinction between OOD and ID data. GOOD-D (Liu et al., 2023) employs hierarchical contrastive learning with structural features exposed to ID samples during training. HGOE (He et al., 2024) integrates realistic graph data from external distribution while synthesizing internal outliers within ID data. However, previous efforts primarily focus on utilizing topological structure similarities to identify structural patterns for OOD detection, overlooking the underlying contextual affinities within the same data distribution, which results in suboptimal performance.

**Concordance Learning.** Concordance learning, which promotes consistent predictions on unlabeled data and its corresponding augmented versions, is a crucial component of semi-supervised learning approaches heavily used in image processing (Sajjadi et al., 2016; Zhang et al., 2023). Recently, motivated by its fascinating capability in capturing self-supervised information, concordance learning is also widely introduced in graph data. For example, TGNN (Ju et al., 2022) leverages KL divergence to promote consistency between the similarity distributions of each unlabeled graph and the labeled graphs stored in the memory bank. HEAL (Ju et al., 2024a) utilizes concordance learning to mutually supervise and reinforce the hypergraph view and the line graph view. To the best of our knowledge, we are the first to apply concordance learning to the OOD task, aiming to explore the semantic structure within ID graphs in contextually affinitive space.

## 3 THE PROPOSED `CARE`

### 3.1 PROBLEM DEFINITION.

**Notations.** Let $G = \{\mathcal{V}, \mathcal{E}, \mathbf{X}\}$ be an attributed and undirected graph, where $\mathcal{V}$ and $\mathcal{E} \in \mathcal{V} \times \mathcal{V}$ represents the node and edge set with $n$ nodes and $m$ edges, respectively. We denote the $\mathbf{X} \in \mathbb{R}^{n \times d'}$ as the node feature attribute matrix, where each row $\mathbf{x}_v$ represents the feature vector of node $v$ with feature dimension $d'$. The graph structure can also represented by an adjacency matrix $\mathbf{A} \in \{0, 1\}^{|\mathcal{V}| \times |\mathcal{V}|}$, where each entry $\mathbf{A}_{uv} = 1$ if edge $(u, v) \in \mathcal{E}$ and vice versa.

**Graph-level OOD Detection.** Given a training dataset $\mathcal{D}_{train} = \{G_1, \ldots, G_N\}$, where each graph $G_i = \{\mathcal{V}_i, \mathcal{E}_i, \mathbf{X}_i\}$ is sampled from a specific distribution $\mathbb{P}^{in}$, we have an ID dataset $\mathcal{D}_{test}^{in}$ at test time. We assume there is also an OOD dataset $\mathcal{D}_{test}^{out}$ where graphs are sampled from an OOD distribution $\mathbb{P}^{out}$ and the test dataset can be defined as $\mathcal{D}_{test} = \mathcal{D}_{test}^{in} \cup \mathcal{D}_{test}^{out}$. The objective of graph-level OOD detection is to train a detection model $f(\cdot)$ on $\mathcal{D}_{train}$ and then predict the source distribution of a graph sample $G \in \mathcal{D}_{test}$ (i.e., $\mathbb{P}^{in}$ or $\mathbb{P}^{out}$). In practice, $f(\cdot)$ can be a scoring function, which can be defined as:

$$\text{Detection label} = \begin{cases} 1 \text{ (OOD)}, & \text{if } f(G') \geqslant \eta \\ 0 \text{ (ID)}, & \text{if } f(G') < \eta \end{cases} \tag{1}$$

where $\eta$ denotes the threshold. It should be noted that each graph from both $\mathbb{P}^{in}$ and $\mathbb{P}^{out}$ may fall into multiple categories, adding complexity to the detection task.

**Graph Neural Networks.** Graph Neural Networks have recently become a prominent approach for learning from graph-structured data. Typically, GNNs operate through a series of message-passing

Figure 2: Overview of the proposed `CARE`. Given ID graphs, we perform message passing on the original and augmented views, and retrieve the nearest neighbors in contextually affinitive space. Then, we apply twin concordance learning to achieve discriminative boundary. Finally, we apply decision boundary compression $\mathcal{L}_r$ to enhance the representation for OOD detection.

layers, where each layer refines the node representations by aggregating and combining features from neighboring nodes. The process can be:

$$\mathbf{h}_v^{(l)} = \mathcal{M}^{(l)}(\mathbf{h}_v^{(l-1)}, \mathcal{A}^{(l)}(\{\mathbf{h}_u^{(l-1)}\}_{u \in \mathcal{N}(v)})), \tag{2}$$

where $\mathbf{h}_v^{(l)}$ denotes the refined embedding of node $v$ at $l$-th layer and we initialize $\mathbf{h}_v^0$ as $\mathbf{x}_v$ in practice. $\mathcal{N}(v)$ is the neighbor set of $v$. $\mathcal{A}(\cdot)$ and $\mathcal{M}(\cdot)$ are the two basic functions that aggregate and combine information from neighbors at the previous layer. After $L$ layers of message-passing, a readout function can be applied to produce a graph-level representation, defined as:

$$\mathbf{h}_G = \text{READOUT}(\{\mathbf{h}_v^{(L)}\}_{v \in \mathcal{V}}), \tag{3}$$

where $\mathbf{h}_G$ is the representation of $G$ and $\text{READOUT}(\cdot)$ can be implemented via different types of pooling functions (i.e., mean and sum) in a permutation-invariant manner.

## 3.2 FRAMEWORK OVERVIEW

The basic idea of our `CARE` is to further consider the contextual relations among the graph data for OOD detection. As shown in Figure 2, our framework comprises three modules. Given the input graph, we retrieve the nearest neighbors by considering contextual relations among the whole batch data. Then, based on the transformed contextually affinitive space, we treat the input graph and its augmentation as two views and group the graph into semantically meaningful clusters. Neighborhood-based consistency is encouraged to achieve both instance- and contextual-aware concordance. This ensures the model produces a consistent and discriminative boundary for different underlying semantics within ID graphs. Finally, we progressively compress the decision boundary within ID samples to achieve more effective OOD detection.

## 3.3 CONTEXTUAL AFFINITY EXPLORATION FOR SEMANTIC STRUCTURE UNDERSTANDING

To capture the contextual relations of the graph, we transform the graph data points into a contextually affinitive space and redefine the graph similarity to retrieve the nearest neighbors. Formally, given a batch of input graph $\mathcal{B} = [G_1, \ldots, G_{|\mathcal{B}|}]$, we leverage the GNN-based encoder to get the graph representation $\mathcal{H} = [\mathbf{h}_1, \ldots, \mathbf{h}_{|\mathcal{B}|}]$. Then, we construct the nearest neighbor enhanced affinity graph to explore the high-order contextual information between graphs, assuming that the more high-order contextual information two graphs share, the more similar the corresponding graph data points are in the transformed space.

**Neighborhood-based Affinity Construction.** For each graph in the batch, we retrieve its nearest neighbors using the dot product distance metric, namely $\langle \mathcal{H}, \mathcal{H}^{\mathrm{T}} \rangle$. In this way, the online reciprocal contextual matrix $\mathbf{R} \in \mathbb{R}^{|\mathcal{B}| \times |\mathcal{B}|}$ are estimated on the resulting $k$-NN graph, defined as:

$$\mathbf{R}(i,j) = \begin{cases} 1, & \text{if } \mathbf{h}_i \in \mathcal{N}(\mathbf{h}_j, k_1) \wedge \mathbf{h}_j \in \mathcal{N}(\mathbf{h}_i, k_1) \\ 0, & \text{if } \mathbf{h}_i \notin \mathcal{N}(\mathbf{h}_j, k_1) \wedge \mathbf{h}_j \notin \mathcal{N}(\mathbf{h}_i, k_1) \\ 0.5, & \text{otherwise} \end{cases}, \tag{4}$$

where $\mathbf{h}_j \in \mathcal{N}(h_i, k_1)$ denote $\mathbf{h}_j$ belongs to the $k_1$ nearest neighbors of $\mathbf{h}_j$ within $\mathcal{B}$. Based on the contextual matrix, we can transform the encoded graph representation into a contextually refined space, and the online global relation graph $G_\mathcal{B} = \{\mathcal{B}, \mathcal{E}_\mathcal{B}, \mathbf{X}_\mathcal{B}\}$ can be formulated as:

$$
\begin{aligned}
\mathcal{E}_\mathcal{B} &= \{w_{ij} | \mathbf{h}_j \in \mathcal{N}(\mathbf{h}_i, k_2), i \in \{1, \dots, |\mathcal{B}|\}\}, \\
\mathbf{X}_\mathcal{B} &= \{x_i | x_i = [\mathbf{R}_{i1}, \dots, \mathbf{R}_{i|\mathcal{B}|}], i \in \{1, \dots, |\mathcal{B}|\})]\}
\end{aligned}
\tag{5}
$$

where $w_{ij} = \langle \mathbf{h}_i, \mathbf{h}_j \rangle$ denotes the edge weight and $\mathbf{h}_j \in \mathcal{N}(\mathbf{h}_i, k_2)$ indicate $\mathbf{h}_j$ is among the $k_2$ nearest neighbors of $\mathbf{h}_j$ in $\mathcal{B}$. In practice, we set $k_2$ far less than $k_1$.

**High-order Affinity Enhancement.** After the construction of the global contextual affinity graph, we leverage a meta-GNN with the message-passing mechanism to further obtain the high-order contextual relations among the graph, which can be computed as:

$$
\mathbf{z}_i^{(l)} = \mathcal{M}^{(l)}(\mathbf{z}_v^{(l-1)}, \mathcal{A}^{(l)}(\{w_{ij}^\alpha \mathbf{z}_j^{(l-1)} | w_{ij} \in \mathcal{E}_\mathcal{B}\})),
\tag{6}
$$

where $w_{ij} \in \mathcal{E}_\mathcal{B}$ ensures that the nearest $k_2$ confident neighbors are selected to explore high-order relations, $\alpha$ is the hyper-parameter for weighted query expansion (Radenović et al., 2018; Yu et al., 2023). We initialize the embedding of $G_i$ within the contextually affinitive space as $\mathbf{z}_i^{(0)} = \mathbf{x}_i \in \mathbf{X}_\mathcal{B}$. After $L$ layers propagation, the refined contextual feature can be $\mathbf{z}_i^L$, which explicitly contains the high-order affinitive information between graphs. Therefore, the affinitive neighborhood can be:

$$
\mathcal{N}^a(i, k) = \{G_j | \mathbf{z}_j^{(L)} \in \mathcal{N}(\mathbf{z}_i^{(L)}, k)\}.
\tag{7}
$$

This process ensures more informative $k$ reciprocal neighbors in contextually affinitive space.

### 3.4 Twin Concordance Learning for Representation Learning

Having acquired contextually affinitive neighbors, we aim to establish the decision boundary in a self-supervised manner for graph OOD detection. Although contrastive learning has proven its effectiveness in capturing common patterns of ID graphs (Liu et al., 2023), we argue that most of the previous methods treat all other ID graphs as negative samples equally. This is contradictory to the fact that some so-called "negative" graphs can be similar or even in the same semantic class as the input graph. To tackle this issue, we compute the relational similarity distribution between the input graph and its contextually affinitive neighbors, and propose an perturbation- and contextual-aware concordance approach to guide the model in learning consistent semantics for ID graph data.

**Perturbation-aware Concordance.** Perturbation-aware concordance encourages the input graph from different views to maintain the same relation distribution with the contextually affinitive neighbors. Specifically, given the input graph $G_i$, we randomly employ one of four fundamental graph data augmentation strategies (You et al., 2020), namely 1) *Node Dropping* 2) *Edge Perturbation* 3) *Attribute Masking* and 4) *Subgraph*, to generate another view of the graph $G_i'$ while preserving their intrinsic structural and attribute information. Then, we encode the original graph and its augmented view by the GNN-based encoder to get $\mathbf{h}_i$ and $\mathbf{h}_i'$, respectively. Here we seek to enhance the consistency of relation distribution for two views. Therefore, we denote the similarity distribution between the original graph $G_i$ and its contextually affinitive neighbors $G_j \in \mathcal{N}^\alpha(i, k)$ as:

$$
Q(i) = \frac{\exp\left(\langle \mathbf{h}_i, \mathbf{h}_j \rangle\right)/\tau}{\sum_{G_j \in \mathcal{N}^\alpha(i,k)} \exp\left(\langle \mathbf{h}_i, \mathbf{h}_j \rangle\right)/\tau},
\tag{8}
$$

where $\mathbf{h}_j$ denotes the neighbor embedding and $\tau$ is a temperature hyper-parameter. And $Q(i)$ can be seen as the probability that $G_i$ selects $G_j$ as its matching context from affinitive neighbors. Similarly, we can also calculate the similarity distribution between the augmented view $G_i'$ and affinitive neighbors $G_j \in \mathcal{N}^\alpha(i, k)$:

$$
P(i) = \frac{\exp\left(\langle \mathbf{h}_i', \mathbf{h}_j \rangle\right)/\tau}{\sum_{G_j \in \mathcal{N}^\alpha(i,k)} \exp\left(\langle \mathbf{h}_i', \mathbf{h}_j \rangle\right)/\tau}.
\tag{9}
$$

For each graph $G_i$, we impose the consistency between the probability distribution $P$ and $Q$, and the loss function can be calculated as:

$$
\mathcal{L}_{ins} = \frac{1}{2N} \sum_{G_i} (D_{KL}(P(i) \| Q(i)) + D_{KL}(Q(i) \| P(i))).
\tag{10}
$$

**Contextual-aware Concordance.** Compared to maintain graph data correlations, contextual-aware concordance regards the local neighborhood as a semantic group and pushes the model to learn a consistent and discriminative boundary for the ID graph data. To this end, we introduce a clustering function $\phi$ that is parameterized by a neural network with softmax on the output to classify graph $G_i$ and its contextually affinitive neighbors $G_j \in \mathcal{N}^\alpha(i,k)$ together, namely defined as $\phi(\mathbf{h}_i) \in [0,1]^C$, which performs a soft assignment over the clusters $\mathcal{C} = \{1, \ldots, C\}$. And the contextual-aware consistency loss can be defined as:

$$\mathcal{L}_{con} = -\frac{1}{N}\sum_{G_i}\sum_{G_j}\log\langle\phi(\mathbf{h}_i), \phi(\mathbf{h}_j)\rangle + \lambda\sum_{c \in \mathcal{C}}\phi'_c \log \phi'_c, \tag{11}$$

with

$$\phi'_c = \sum_{G_i}\phi_c(\mathbf{h}_i). \tag{12}$$

Note that the first term of loss imposes that each graph $G_i$ and its neighboring graph $\mathcal{N}^\alpha(i,k)$ make the same semantic group prediction. This term reaches its maximum when the predictions are one-hot (confident) and are assigned to the same cluster (consistent). Meanwhile, we introduce an entropy loss as the second term to avoid trivial solutions of $\phi$, which assigns all samples to a single cluster. Here $\phi_c(\mathbf{h}_i)$ denote the probability of $G_i$ assigned to cluster $c$.

### 3.5 DECISION BOUNDARY COMPRESSION FOR REPRESENTATION ENHANCEMENT

With the learned cluster structure of the ID graph data serving as the decision boundary, we focus on capturing the holistic of the semantics. This emphasizes that farther samples located at the cluster boundary still retain separability. Inspired by the silhouette score (Rousseeuw, 1987), which measures the ratio of the mean intra-cluster distance $d^I$ to the mean nearest-cluster distance $d^N$ in an offline setting, we introduce an approximate online version to compress the decision boundary of the ID graph data:

$$d_i^I = \|\mathbf{h}_i - \boldsymbol{\mu}_{\phi(\mathbf{h}_i)}\|_2, \quad d_i^N = \|\mathbf{h}_i - \boldsymbol{\mu}_{\phi'(\mathbf{h}_i)}\|_2, \tag{13}$$

where

$$\phi'(\mathbf{h}_i) = \underset{c \in \{1,\ldots,C\}\setminus\{\phi(\mathbf{h}_i)\}}{\arg\min}\|\mathbf{h}_i - \boldsymbol{\mu}_c\|, \tag{14}$$

with $\boldsymbol{\mu}_c$ denote the center of cluster $c$ obtained from last epoch. The boundary compression loss can be defined as:

$$\mathcal{L}_r = \sum_{G_i}r_i, \; r_i = 1 - \frac{d_i^N - d_i^I}{\max(d_i^I, d_i^N)}, \tag{15}$$

where $r_i$ denotes the compression ratio and a larger $r_i$ indicates that the graph sample $G_i$ is more likely located at the boundary. Therefore, we minimize the compression ratio to impose the ID graph sample close to its cluster center. More theoretical discussion on the proposed compression strategy is provided in Appendix C.

### 3.6 SUMMARIZATION

The overall objective is a combination of perturbation-aware consistency regularization loss, contextual-aware consistency regularization loss and decision boundary compression loss. Formally, the final loss of our `CARE` is:

$$\mathcal{L} = \mathcal{L}_{ins} + \beta\mathcal{L}_{con} + \gamma\mathcal{L}_r, \tag{16}$$

where $\beta$ and $\gamma$ are weight coefficients used to control their respective contributions. During the test phase, we directly employ boundary compression ratio $r$ as the OOD score of the test graph sample. The learning procedure of our proposed `CARE` is illustrated in Algorithm 1 in Appendix B.

## 4 EXPERIMENTS

### 4.1 EXPERIMENT SETUP

**Datasets.** We choose 5 pairs of datasets from the OGB dataset (Hu et al., 2020) and 5 pairs from the TU dataset (Morris et al., 2020). Each pair shares the same field and similar features yet shows

Table 1: Comparison of OOD detection methods using AUC scores, reported as mean ± std. The top-performing and second-best results are highlighted in **bold** and underline, respectively.

| ID Dataset
OOD Dataset | BZR
COX2 | PTC-MR
MUTAG | AIDS
DHFR | ENZYMES
PROTEIN | IMDB-M
IMDB-B | Tox21
SIDER | FreeSolv
ToxCast | BBBP
BACE | ClinTox
LIPO | Esol
MUV |
|---|---|---|---|---|---|---|---|---|---|---|
| PK-LOF | 42.22±8.39 | 51.04±6.04 | 50.15±3.29 | 50.47±2.87 | 48.03±2.53 | 51.33±1.81 | 49.16±3.70 | 53.10±2.07 | 50.00±2.17 | 50.82±1.48 |
| PK-OCSVM | 42.55±8.26 | 49.71±6.58 | 50.17±3.30 | 50.46±2.78 | 48.07±2.41 | 51.33±1.81 | 48.82±3.29 | 53.05±2.10 | 50.06±2.19 | 51.00±1.33 |
| PK-iF | 51.46±1.62 | 54.29±4.33 | 51.10±1.43 | 51.67±2.69 | 50.67±2.47 | 49.87±0.82 | 52.28±1.87 | 51.47±1.33 | 50.81±1.10 | 50.85±3.51 |
| WL-LOF | 48.99±6.20 | 53.31±8.98 | 50.77±2.87 | 52.66±2.47 | 52.28±4.50 | 51.92±1.58 | 51.47±4.23 | 52.80±1.91 | 51.29±3.40 | 51.26±1.31 |
| WL-OCSVM | 49.16±4.51 | 53.31±7.57 | 50.98±2.71 | 51.77±2.21 | 51.38±2.39 | 51.08±1.46 | 50.38±3.81 | 52.85±2.00 | 50.77±3.69 | 50.97±1.65 |
| WL-iF | 50.24±2.49 | 51.43±2.02 | 50.10±0.44 | 51.17±2.01 | 51.07±2.25 | 50.25±0.96 | 52.60±2.38 | 50.78±0.75 | 50.41±2.17 | 50.61±1.96 |
| InfoGraph-iF | 63.17±9.74 | 51.43±5.19 | 93.10±1.35 | 60.00±1.83 | 58.73±1.96 | 56.28±0.81 | 56.92±1.69 | 53.68±2.90 | 48.51±1.87 | 54.16±5.14 |
| InfoGraph-MD | 86.14±6.77 | 50.79±8.49 | 69.02±11.67 | 55.25±3.51 | 81.38±1.14 | 59.97±2.06 | 58.05±5.46 | 70.49±4.63 | 48.12±5.72 | 77.57±1.69 |
| GraphCL-iF | 60.00±3.81 | 50.86±4.30 | 92.90±1.21 | 61.33±2.27 | 59.67±1.65 | 56.81±0.97 | 55.55±2.71 | 59.41±3.58 | 47.84±0.92 | 62.12±4.01 |
| GraphCL-MD | 83.64±6.00 | 73.03±2.38 | 93.75±2.13 | 52.87±6.11 | 79.09±2.73 | 58.30±1.52 | 60.31±5.24 | 75.72±1.54 | 51.58±3.64 | 78.73±1.40 |
| OCGIN | 76.66±4.17 | 80.38±6.84 | 86.01±6.59 | 57.65±2.96 | 67.93±3.86 | 46.09±1.66 | 59.60±4.78 | 61.21±8.12 | 49.13±4.13 | 54.04±5.50 |
| GLocalKD | 75.75±5.99 | 70.63±3.54 | 93.67±1.24 | 57.18±2.03 | 78.25±4.35 | 66.28±0.98 | 64.82±3.31 | 73.15±1.26 | 55.71±3.81 | 86.83±2.35 |
| GOOD-D | 94.99±2.25 | 81.21±2.65 | 99.07±0.40 | 61.84±1.94 | 79.94±1.09 | 66.50±1.35 | 80.13±3.43 | 82.91±2.58 | 69.18±3.61 | 91.52±0.70 |
| HGOE | 95.00±2.70 | 82.06±1.63 | 99.28±0.34 | 64.44±2.19 | 81.74±2.25 | 68.24±0.60 | **82.89±2.33** | 83.46±1.79 | 70.09±1.52 | 92.64±2.44 |
| CGOD | 95.47±3.85 | 81.42±2.04 | 98.17±0.28 | 61.52±1.62 | 79.02±3.80 | 69.10±1.58 | 81.72±1.07 | 81.75±1.43 | 65.13±2.61 | 89.68±3.02 |
| **CARE** | **96.32±1.16** | **82.60±2.14** | **99.42±0.16** | **64.86±2.48** | **82.44±1.76** | **70.69±0.42** | 82.00±1.16 | **84.21±1.07** | **71.31±2.10** | **93.11±1.06** |

Table 2: Comparison of anomaly detection methods using AUC scores, reported as mean ± std. The top-performing and second best results are highlighted in **bold** and underline, respectively.

| Method | PK-OCSVM | PK-iF | WL-OCSVM | WL-iF | InfoGraph-iF | GraphCL-iF | OCGIN | GLocalKD | GOOD-D | HGOE | CARE |
|---|---|---|---|---|---|---|---|---|---|---|---|
| PROTEINS-full | 50.49±4.92 | 60.70±2.55 | 51.35±4.35 | 61.36±2.54 | 57.47±3.03 | 60.18±2.53 | 70.89±2.44 | 77.30±5.15 | 71.97±3.86 | 73.13±0.46 | **77.82±3.50** |
| ENZYMES | 53.67±2.66 | 51.30±2.01 | 55.24±2.66 | 51.60±3.81 | 53.80±4.50 | 53.60±4.88 | 58.75±5.98 | 61.39±8.81 | 63.90±3.69 | 67.28±0.99 | **68.08±5.78** |
| AIDS | 50.79±4.30 | 51.84±2.87 | 50.12±3.43 | 61.13±0.71 | 70.19±5.03 | 79.72±3.98 | 78.16±3.05 | 93.27±4.19 | 97.28±0.69 | 97.84±0.55 | **98.56±0.49** |
| DHFR | 47.91±3.76 | 52.11±3.96 | 50.24±3.13 | 50.29±2.77 | 52.68±3.21 | 51.10±2.35 | 49.23±3.05 | 56.71±3.57 | 62.67±3.11 | 64.39±0.68 | **65.95±4.20** |
| BZR | 46.85±5.31 | 55.32±6.18 | 50.56±5.87 | 52.46±3.30 | 63.31±8.52 | 60.24±5.37 | 65.91±1.47 | 69.42±7.78 | 75.16±5.15 | **80.54±1.35** | 77.02±6.22 |
| COX2 | 50.27±7.91 | 50.05±2.06 | 49.86±7.43 | 50.27±0.34 | 53.36±8.86 | 52.01±3.17 | 53.58±5.05 | 59.37±12.67 | 62.65±8.14 | **69.52±2.68** | 65.70±5.05 |
| DD | 48.30±3.98 | 71.32±2.41 | 47.99±4.09 | 70.31±1.09 | 55.80±1.77 | 59.32±3.92 | 72.27±1.83 | 80.12±5.24 | 73.25±3.19 | 76.95±2.24 | **81.91±2.81** |
| NCI1 | 49.90±1.18 | 50.58±1.38 | 50.63±1.22 | 50.74±1.70 | 50.10±0.87 | 49.88±0.53 | **71.98±1.21** | 68.48±2.39 | 61.12±2.21 | 65.82±1.43 | 67.06±1.94 |
| IMDB-B | 50.75±3.10 | 50.80±3.17 | 54.08±5.19 | 50.20±0.40 | 56.50±3.58 | 56.50±4.90 | 60.19±8.90 | 52.09±3.41 | 65.88±0.75 | 69.82±1.37 | **73.14±4.07** |
| REDDIT-B | 45.68±2.24 | 46.72±3.42 | 49.31±2.33 | 48.26±0.32 | 68.50±5.56 | 71.80±4.38 | 75.93±8.65 | 77.85±2.62 | 88.67±1.24 | 89.41±1.21 | **89.92±1.25** |
| HSE | 57.02±8.42 | 56.87±10.51 | 62.72±10.13 | 53.02±5.12 | 53.56±1.44 | 51.18±2.71 | 64.84±4.70 | 59.48±1.44 | 69.65±2.14 | **74.50±3.73** | 66.50±1.21 |
| MMP | 46.65±6.31 | 50.06±3.73 | 55.24±3.26 | 52.68±3.34 | 54.59±2.01 | 54.54±1.86 | 71.23±0.16 | 67.84±0.59 | 70.51±1.56 | **71.94±0.54** | 70.27±1.30 |
| p53 | 46.74±4.88 | 50.69±2.02 | 54.59±4.46 | 50.85±2.16 | 52.66±1.95 | 53.29±2.32 | 58.50±0.37 | 64.20±0.81 | 62.99±1.55 | 64.70±1.16 | **66.41±0.35** |
| PPAR-gamma | 53.94±6.94 | 45.51±2.58 | 57.91±6.13 | 49.60±0.22 | 51.40±2.53 | 50.30±1.56 | 71.19±4.28 | 64.59±0.67 | 67.34±1.71 | 71.92±4.17 | **72.10±0.91** |
| Avg. Rank | 9.7 | 8.8 | 8.1 | 8.6 | 7.4 | 7.6 | 4.6 | 4.1 | 3.4 | 2.0 | **1.6** |

distribution shifts. For each pair, 90% of samples from one dataset are used as ID samples for training, while the remaining samples, along with an equivalent number OOD samples from the other dataset, are used for testing. To evaluate performance in anomaly detection, we select 14 datasets from the TU dataset (Morris et al., 2020). Consistent with prior research (Ma et al., 2022; Liu et al., 2023), samples from the majority class are considered as normal data for training, while samples from the minority class or real anomalous class are treated as anomalies.

**Baselines.** We compare CARE against 15 baselines. Specifically, these approaches can be categorized into three groups: **non-deep two-step methods** (i.e., *Weisfeiler-Lehman kernel (WL)* (Shervashidze et al., 2011) and *propagation kernel (PK)* (Neumann et al., 2016) are used as feature extractors, while *local outlier factor (LOF)* (Breunig et al., 2000), *one-class SVM (OCSVM)* (Manevitz & Yousef, 2001), and *isolation forest (iF)* (Liu et al., 2008) are used as detectors), **deep two-step methods** (i.e., *InfoGraph* (Sun et al., 2020) and *GraphCL* (You et al., 2020) are used as feature extractors, while *iF* and *Mahalanobis distance-based (MD)* are used as detectors), and **end-to-end methods** (i.e., *OCGIN* (Zhao & Akoglu, 2023), *GLocalKD* (Ma et al., 2022), *GOOD-D* (Liu et al., 2023), *HGOE* (He et al., 2024), and CGOD (Lin et al., 2025)).

**Evaluation and Implementation.** Following previous research (Liu et al., 2023; He et al., 2024), we use area under receiver operating characteristic curve (AUC) to measure the performance of our model. All experiments are conducted 5 times, and we report the mean AUC along with the standard deviation. More experimental details are provided in Appendix D. The code is available at `https://anonymous.4open.science/r/CARE-for-Graph-OOD-Detection-C49C`.

## 4.2 PERFORMANCE COMPARISON

**OOD detection.** Table 1 shows the results of our CARE against all baselines. From the results, we can draw the following observations. ❶ Compared to non-deep two-step methods, deep learning-based approaches consistently deliver better performance by leveraging high-dimensional features and topological information. ❷ End-to-end methods typically outperform two-step detection approaches, suggesting that integrating the feature extractor and detector enhances the refinement of tailored representations. ❸ Our proposed CARE demonstrates superior performance compared to all

Table 3: Ablation study results of `CARE` and its variants in terms of AUC (in percent, mean ± std).

| $\mathcal{L}_{ins}$ | $\mathcal{L}_{con}$ | $\mathcal{L}_r$ | BZR COX2 | PTC-MR MUTAG | AIDS DHFR | ENZYMES PROTEIN | IMDB-M IMDB-B | Tox21 SIDER | FreeSolv ToxCast | BBBP BACE | ClinTox LIPO | Esol MUV |
|---|---|---|---|---|---|---|---|---|---|---|---|---|
| ✓ | ✗ | ✗ | $80.57_{\pm5.15}$ | $79.71_{\pm7.71}$ | $95.64_{\pm1.14}$ | $54.26_{\pm3.10}$ | $75.02_{\pm1.80}$ | $69.01_{\pm0.51}$ | $64.50_{\pm3.12}$ | $73.71_{\pm1.08}$ | $58.06_{\pm2.41}$ | $88.04_{\pm1.81}$ |
| ✗ | ✓ | ✗ | $77.41_{\pm6.38}$ | $76.42_{\pm5.88}$ | $97.10_{\pm2.03}$ | $53.27_{\pm1.67}$ | $74.08_{\pm2.44}$ | $68.65_{\pm0.59}$ | $67.54_{\pm5.61}$ | $76.42_{\pm2.12}$ | $58.22_{\pm2.90}$ | $88.28_{\pm1.43}$ |
| ✗ | ✗ | ✓ | $94.52_{\pm1.47}$ | $79.11_{\pm4.81}$ | $98.06_{\pm0.32}$ | $58.86_{\pm3.28}$ | $79.29_{\pm1.57}$ | $69.13_{\pm0.85}$ | $69.42_{\pm4.05}$ | $79.68_{\pm0.87}$ | $62.57_{\pm1.58}$ | $90.69_{\pm0.71}$ |
| ✓ | ✓ | ✗ | $80.96_{\pm5.88}$ | $75.98_{\pm6.34}$ | $97.01_{\pm1.93}$ | $59.57_{\pm3.25}$ | $79.67_{\pm3.29}$ | $69.40_{\pm0.78}$ | $72.38_{\pm4.74}$ | $81.56_{\pm1.17}$ | $69.35_{\pm6.23}$ | $91.25_{\pm1.39}$ |
| ✓ | ✗ | ✓ | $95.73_{\pm1.88}$ | $79.66_{\pm4.91}$ | $\underline{98.92_{\pm0.27}}$ | $61.25_{\pm2.70}$ | $\underline{81.54_{\pm2.43}}$ | $69.95_{\pm0.79}$ | $\underline{78.15_{\pm4.32}}$ | $83.76_{\pm1.26}$ | $67.93_{\pm4.90}$ | $\underline{92.66_{\pm0.83}}$ |
| ✗ | ✓ | ✓ | $\underline{95.78_{\pm1.43}}$ | $\underline{82.48_{\pm2.56}}$ | $98.90_{\pm0.39}$ | $\underline{61.76_{\pm5.82}}$ | $80.79_{\pm3.13}$ | $\underline{69.78_{\pm0.55}}$ | $76.37_{\pm4.07}$ | $65.80_{\pm1.18}$ | $\underline{70.20_{\pm1.01}}$ | $92.53_{\pm1.42}$ |
| ✓ | ✓ | ✓ | $\mathbf{96.32_{\pm1.16}}$ | $\mathbf{82.60_{\pm2.14}}$ | $\mathbf{99.42_{\pm0.16}}$ | $\mathbf{64.86_{\pm2.48}}$ | $\mathbf{82.44_{\pm1.76}}$ | $\mathbf{70.69_{\pm0.42}}$ | $\mathbf{82.00_{\pm1.16}}$ | $\mathbf{84.21_{\pm1.07}}$ | $\mathbf{71.31_{\pm2.10}}$ | $\mathbf{93.11_{\pm1.06}}$ |

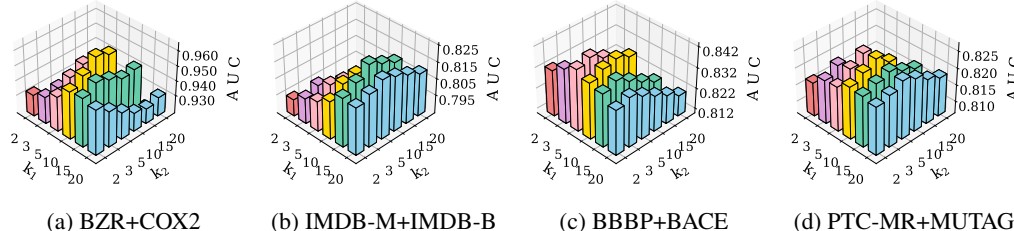

(a) BZR+COX2     (b) IMDB-M+IMDB-B     (c) BBBP+BACE     (d) PTC-MR+MUTAG

Figure 3: Performance comparison w.r.t. different values of $k_1$ and $k_2$ for contextual affinity graph.

baseline methods on most datasets. Notably, it achieves average AUC improvements of $1.16\%$ and $2.65\%$ relative to two baselines HGOE and GOOD-D, respectively. We attribute enhancement of OOD detection to two reasons. ❶ The utilization of contextual affinity enhances the understanding of the semantic structure within the ID dataset, fostering greater compactness in the underlying feature space. ❷ The introduction of twin affinity-aware concordance learning further reveals the preserved semantic structure, enabling robust and discriminative graph representations.

**Anomaly detection.** Following the benchmark established in (Ma et al., 2022), we conduct anomaly detection experiments to investigate whether `CARE` can generalize to this task, with results presented in Table 2. Our findings indicate that: ❶ Consistent with the results in Table 1, end-to-end approaches outperform two-step approaches, while non-deep two-step methods demonstrate the weakest performance. ❷ `CARE` shows strong competitiveness in anomaly detection, highlighting its promising capability to capture the contextual patterns of normal data through the leveraging of semantic neighborhoods.

### 4.3 ABLATION STUDY

In this section, we conduct ablation experiments to illustrate the impact of the key components. We perform experiments on all combinations of loss functions. From the results shown in Table 3, we conclude that: ❶ Each component significantly contributes to overall performance, with decision boundary compression having the greatest impact due to its contribution in refining discriminative boundary. ❷ The combination of two components typically yields better results than simply using individual component. ❸ `CARE` with all components achieves the highest performance. These observations highlight the effectiveness of jointly utilizing the carefully designed components.

### 4.4 PARAMETER ANALYSIS

**Analysis of contextual affinity graph construction.** We first analyze the optimal value of $k_1$ and $k_2$ in the construction of affinity graph in the range of $\{2, 3, 5, 10, 15, 20\}$ (note that $k_2 < k_1$). As shown in Figure 3, we have following conclusions. ❶ $k_2$ that is too small or too close to $k_1$ can lead to performance degradation due to insufficient information integration or the introduction of noise. ❷ The optimal values of both $k_1$ and $k_2$ vary across datasets. We conjecture that the best selections are closely related to the underlying properties of ID datasets, such as the number of categories or typical patterns. Based on our experience, we recommend that $k_1 \in \{5, 10, 15\}$ and $k_2 \in \{3, 5\}$ usually yield promising performance.

**Analysis of cluster numbers.** We then study the sensitivity of `CARE` w.r.t. the cluster number $C$ in the range of $\{2, 3, 5, 10, 15, 20\}$. As shown in Figure 4, we discover that the optimal value of $C$ varies across dataset pairs, indicating its inherent correlations with the size of datasets. For example,

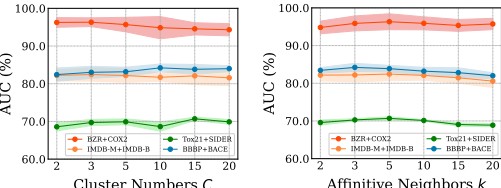 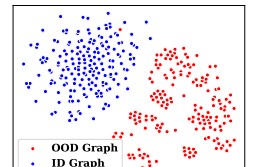 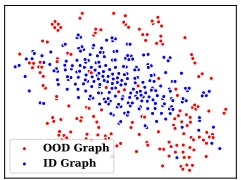

Figure 4: Performance comparison w.r.t. different numbers of clusters $C$ (left) and affinitive neighbors $k$ (right).

Figure 5: t-SNE visualization of the graph embeddings of two dataset pairs: AIDS+DHFR (left) and BBBP+BACE (right).

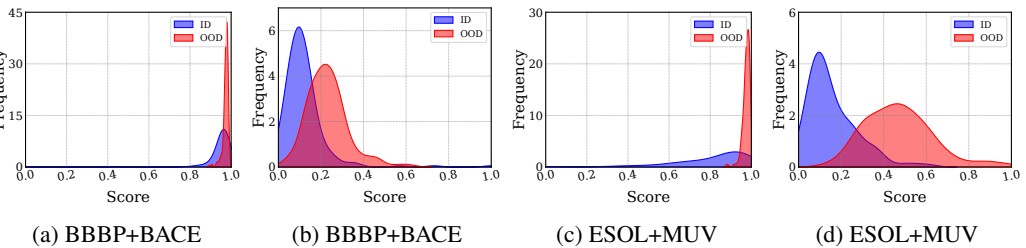

| (a) BBBP+BACE | (b) BBBP+BACE | (c) ESOL+MUV | (d) ESOL+MUV |

Figure 6: The score distributions before training (a, c) and after training (b, d) with our method.

the best performance is achieved with $C = 15$ in Tox21+SIDER, while a smaller $C$ (i.e., $C = 3$) is preferred by BZR+COX2. Nevertheless, `CARE` is not sensitive to $C$.

**Analysis of reciprocal neighbor numbers.** We further investigate the sensitivity of the number of reciprocal neighbor $k$ in the range of $\{2, 3, 5, 10, 15, 20\}$. From the Figure 4, we conclude that moderately incorporating affinitive neighbors enhances overall performance by effectively leveraging higher-order contextual information. However, excessive incorporation of reciprocal neighbors may introduce noise from false positive samples, resulting in a slight drop in performance. Overall, the performance remains relatively stable, with the best results typically observed when $k = 3, 5$.

## 4.5 CASE STUDY

**Visualization on learned embeddings.** We use 2D t-distributed stochastic neighbor embedding (t-SNE) to visualize the learned embeddings of our model. As shown in Figure 5, on the one hand, the ID samples are well concentrated into a cluster with distinct boundary, exhibiting tight within-cluster aggregation. On the other hand, the OOD samples are positioned far from the cluster center of the ID samples, highlighting clearer distinctions.

**Visualization on score distributions.** To intuitively evaluate the discriminative power of the model, we plotted the score distributions for the ID and OOD test samples. The visualization in Figure 6 reveals that our model clearly separates the peaks of the ID samples from the OOD samples with relatively less overlap area, which yields a promising boundary for distinguishing between the ID and the OOD samples. This observation further demonstrates the effectiveness of `CARE`.

## 5 CONCLUSION

In this paper, we introduce a novel approach called `CARE` for graph OOD detection, emphasizing semantic structure—an aspect often overlooked in previous research and equally important as topological structure. `CARE` first constructs a contextual affinity graph, which is processed by a meta-graph neural network to uncover the underlying semantic structure via high-order affinity. Subsequently, we leverage twin concordance learning to foster robust and discriminative graph representations. Finally, we propose a compression strategy to achieve refined separation between ID and OOD samples. Our extensive experiments on 10 real-world datasets demonstrate that `CARE` yields significant performance across various datasets. A current limitation, however, is the method's dependence on a static contextual matrix, which may introduce noise. In future work, we plan to investigate an adaptive mechanism for selecting the optimal number of affinitive neighbors and expand the framework to address node-level OOD detection.

## 6 REPRODUCIBILITY STATEMENT

We ensure the reproducibility of our work as follows. The main components of the proposed `CARE` framework are detailed in Sections 3, including all mathematical derivations. Experimental settings, including hyperparameters, hardware specifications, datasets, and the complete training pipeline, are provided in Section 4.1 and Appendix D. An anonymous implementation is available at: https://anonymous.4open.science/r/CARE-for-Graph-OOD-Detection-C49C.

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

## A  LARGE LANGUAGE MODEL (LLM) USAGE STATEMENT

We use the LLM as a general-purpose assistant tool. Specifically, the LLM assists in (i) checking grammar and improving clarity of text descriptions, and (ii) suggesting alternative phrasings for some sections. No parts of the paper are generated entirely by the LLM. All research ideas, experiments, model designs, and results are conceived, implemented, and analyzed solely by the authors. The LLM does not contribute to the development of the methodology, experiments, or analysis presented in this paper. We confirm that the use of the LLM is limited to minor writing support and does not constitute a substantive contribution that would qualify it as a co-author.

## B  TRAINING DETAILS

---

**Algorithm 1** The overall training process of `CARE`

---

**Input**: Training set $\mathcal{D}_{train} = \{G_1, G_2, \ldots, G_N\}$ where graph $G_i = \{\mathcal{V}_i, \mathcal{E}_i, X_i\}$ is sampled from ID dataset.

**Output**: OOD scores $r_i$ of each graph $G_i$ in test dataset $\mathcal{D}_{test} = \{G_1, G_2, \ldots, G_N\}$ sampled from ID and OOD dataset.

1: **while** not convergence **do**
2:     Divide the training set into batches $\mathcal{B}$ by random; form two graph views from the original
      graphs and the augmented graphs denoted as $\{G_1, G_2, \ldots, G_\mathcal{B}\}$ and $\{G_1', G_2', \ldots, G_\mathcal{B}'\}$;
3:     Encode the original graphs and the augmented graphs to get graph representations $h_i$ and $h_i'$
      ;                                                        `// Eq. 2-3`
4:     Calculate the dot product distance between each graph in the batch to get contextual matrix
      and construct the online global relation graph $G_\mathcal{B}$ ;              `// Eq. 4-5`
5:     Obtain high-order contextual relations from $G_\mathcal{B}$ to get contextual feature in affinitive space
      and retrieve the nearest neighbors $\mathcal{N}^a(i, k)$ for each graph $G_i$ ;       `// Eq. 6-7`
6:     **for** $G_i, G_i' \in \mathcal{B}$ **do**
7:         Get similarity distribution $Q$ between $G_i$ and $\mathcal{N}^a(i, k)$; get similarity distribution $P$
         between $G_i'$ and $\mathcal{N}^a(i, k)$; calculate the KL divergence between $P$ and $Q$ to obtain $\mathcal{L}_{ins}$ ;
         `// Eq. 8-10`
8:         Perform soft clustering for $G_i$ and $\mathcal{N}^a(i, k)$ to obtain $\mathcal{L}_{con}$ ;         `// Eq. 11`
9:         Calculate the intra-cluster distance $d_i^I$ and nearest-cluster distance $d_i^N$ for $G_i$ to obtain $\mathcal{L}_r$
         ;                                                     `// Eq. 13-15`
10:    **end for**
11:    Update parameters in the model by gradient descent
12: **end while**
13: Obtain $r_i$ for each graph in the test set.
14: **return** $r$

---

## C  THEORETICAL ANALYSIS

In this part, we present a theoretical analysis of the proposed method `CARE`. Let the parameter of the model be denoted as $\theta$. The compression ratio for the $i$-th graph corresponding to the optimal parameter $\theta^*$ is defined as $r_i(\theta^*)$. Suppose there are $C$ clusters in the training set. We consider the graph from the OOD in the test set as belonging to a new cluster. For the $i$-th graph in $\mathcal{D}_{test}^{out}$, let the graph representation be denoted by $\mathbf{h}_i = \mathbf{h}_i(\theta^*)$. With this notation, we define:

$$\boldsymbol{\mu}_{out} = \text{mean} \left\{ h_i(\theta^*) : G_i \in \mathcal{D}_{test}^{out} \right\}. \tag{17}$$

Next, we define the expected compression ratios for the ID and OOD graphs, respectively, as:

$$R_{in} = \mathbb{E}_{in}(r(\theta^*)), \quad R_{out} = \mathbb{E}_{out}(r(\theta^*)). \tag{18}$$

Our next goal is to show that the compression ratio for the OOD graphs is larger than that for the ID graphs, i.e., $R_{out} > R_{in}$. Thus, we can choose a threshold $\eta$ such that if $r(\theta^*) > \eta$, the graph is classified as OOD. The key term in $r(\theta^*)$ is $d_i^N(\theta^*) - d_i^I(\theta^*)$. For simplicity, we denote $d_i^N(\theta^*) - d_i^I(\theta^*)$ as $D$. We will demonstrate that the expected value of this term is smaller for the OOD graphs compared to the ID graphs, i.e., $D_{out} < D_{in}$. Specifically, we have:

$$D_\Delta = \mathbb{E}_\Delta \left( d_i^N(\theta^*) - d_i^I(\theta^*) \right), \Delta \in \{in, out\} \tag{19}$$

**Theorem C.1.** *Assume that*

$$\mathbb{E}_{out} \left( \|\boldsymbol{\mu}_{out} - \boldsymbol{\mu}_{\phi'(h_i)}\|_2 \right) - \mathbb{E}_{out} \left( \|\boldsymbol{\mu}_{out} - \boldsymbol{\mu}_{\phi(h_i)}\|_2 \right)$$

*is sufficiently smaller than* $\mathbb{E}_{in} \left( \|\boldsymbol{\mu}_{\phi(h_i)} - \boldsymbol{\mu}_{\phi'(h_i)}\|_2 \right)$, *where the expectation is taken at the optimal parameter* $\theta^*$. *Then, we have* $D_{out} < D_{in}$.

Theorem C.1 implies that the $D_{out} < D_{in}$, suggesting that the compression ratio for the OOD graphs is larger than that for the ID graphs. Therefore, we can choose a threshold $\eta$ such that if $r > \eta$, the graph is classified as OOD. Note that the analysis in Theorem C.1 is based on the optimal parameter $\theta^*$. In practice, we use the parameter $\hat{\theta}$ that minimizes the empirical loss.

**Theorem C.2.** *Assume that*

$$\sum_{G_i} r_i(\hat{\theta}) - \sum_{G_i} r_i(\theta^*) \le 0. \tag{20}$$

*Then, we have*

$$\mathbb{E}_{in} \left( r(\hat{\theta}) \right) - \mathbb{E}_{in} \left( r(\theta^*) \right) \le C\sqrt{\frac{d_{VC}}{n}}, \tag{21}$$

*where* $C$ *is a positive constant,* $n$ *is the number of training samples, and* $d_{VC}$ *is the Vapnik-Chervonenkis (VC) dimension of the neural network.*

Since the parameter $\hat{\theta}$ is obtained by minimizing the empirical loss, it is reasonable to assume that the empirical compression ratio at $\hat{\theta}$ is less than or equal to that at $\theta^*$. Under this assumption, Theorem C.2 demonstrates that the expected compression ratio at $\hat{\theta}$ is close to that at $\theta^*$. Typically, the VC dimension of the neural network is related to the number of parameters (see Bartlett et al. (2021) for more discussion). Similarly, we have the results for the OOD graphs. The proof of this theorem based on empirical process theory is provided below.

*Proof of Theorem C.1.* We begin by analyzing $D_{in}$:

$$
\begin{aligned}
D_{in} &= \mathbb{E}_{in} \left( d_i^N(\theta^*) - d_i^I(\theta^*) \right) \\
&= \mathbb{E}_{in} \left( \|h_i(\theta^*) - \boldsymbol{\mu}_{\phi(h_i)}\|_2 - \|h_i(\theta^*) - \boldsymbol{\mu}_{\phi'(h_i)}\|_2 \right) \\
&\ge \mathbb{E}_{in} \left( \|h_i(\theta^*) - \boldsymbol{\mu}_{\phi(h_i)}\|_2 - \|h_i(\theta^*) - \boldsymbol{\mu}_{\phi(h_i)}\|_2 \right) \\
&\quad - \mathbb{E}_{in} \left( \|\boldsymbol{\mu}_{\phi(h_i)} - \boldsymbol{\mu}_{\phi'(h_i)}\|_2 \right).
\end{aligned}
$$

Similarly, for $D_{out}$, we have:

$$
\begin{aligned}
D_{out} &= \mathbb{E}_{out} \left( d_i^N(\theta^*) - d_i^I(\theta^*) \right) \\
&= \mathbb{E}_{out} \left( \|h_i(\theta^*) - \boldsymbol{\mu}_{\phi(h_i)}\|_2 - \|h_i(\theta^*) - \boldsymbol{\mu}_{\phi'(h_i)}\|_2 \right) \\
&\le \mathbb{E}_{out} \left( \|\boldsymbol{\mu}_{out} - \boldsymbol{\mu}_{\phi'(h_i)}\|_2 \right) - \mathbb{E}_{out} \left( \|\boldsymbol{\mu}_{out} - \boldsymbol{\mu}_{\phi(h_i)}\|_2 \right) \\
&\quad + 2\mathbb{E}_{out} \left( \|h_i(\theta^*) - \boldsymbol{\mu}_{out}\|_2 \right).
\end{aligned}
$$

By the assumption that

$$\mathbb{E}_{out} \left( \|\boldsymbol{\mu}_{out} - \boldsymbol{\mu}_{\phi'(h_i)}\|_2 \right) - \mathbb{E}_{out} \left( \|\boldsymbol{\mu}_{out} - \boldsymbol{\mu}_{\phi(h_i)}\|_2 \right)$$

is sufficiently smaller than

$$\mathbb{E}_{\text{in}}\left(\|\boldsymbol{\mu}_{\phi(h_i)} - \boldsymbol{\mu}_{\phi'(h_i)}\|_2\right),$$

such that

$$\mathbb{E}_{\text{out}}\left(\|\boldsymbol{\mu}_{\text{out}} - \boldsymbol{\mu}_{\phi'(h_i)}\|_2\right) - \mathbb{E}_{\text{out}}\left(\|\boldsymbol{\mu}_{\text{out}} - \boldsymbol{\mu}_{\phi(h_i)}\|_2\right)$$
$$\leq \mathbb{E}_{\text{in}}\left(\|\boldsymbol{\mu}_{\phi(h_i)} - \boldsymbol{\mu}_{\phi'(h_i)}\|_2\right) + 2\mathbb{E}_{\text{out}}\left(\|h_i(\theta^*) - \boldsymbol{\mu}_{\text{out}}\|_2\right),$$

we conclude that $D_{\text{out}} < D_{\text{in}}$. Thus, the theorem is proved. $\square$

To prove Theorem C.2, we need the following two lemmas.

Define the empirical process $G_n$ as:

$$G_n = \sqrt{n}\frac{1}{n}\sum_{i=1}^{n}\left(r_i(\theta) - \mathbb{E}\left(r_i(\theta)\right)\right),$$

where $r_i(\theta)$ is the compression ratio for the $i$-th graph at parameter $\theta$. Let $\mathcal{F}$ be a class of measurable functions indexed by the parameters of the neural network with envelope function $F \equiv 2$, since $r_i$ is bounded by 2. We define the $L_2$ norm of $G_n$ with respect to $\mathcal{F}$ as:

$$\|G_n\|_{\mathcal{F}} = \sup_{f \in \mathcal{F}}|\langle G_n, f\rangle|,$$

where $\langle \cdot, \cdot \rangle$ denotes the inner product. Define the entropy integral as:

$$J_{[\,]}(\epsilon, \mathcal{F}, L_2(P)) = \int_0^{\epsilon}\sqrt{\log N(\epsilon, \mathcal{F}, L_2(P))}\,d\epsilon,$$

where $N(\epsilon, \mathcal{F}, L_2(P))$ is the covering number of $\mathcal{F}$ with respect to $L_2(P)$. The following lemma provides an upper bound on the expectation of the $L_2$ norm of $G_n$ with respect to $\mathcal{F}$.

**Lemma C.3.** *For any class $\mathcal{F}$ of measurable functions with envelope function $F$,*

$$\mathbb{E}_P^*\|G_n\|_{\mathcal{F}} \lesssim J_{[\,]}\left(\|F\|_{P,2}, \mathcal{F}, L_2(P)\right).$$

To compute the entropy integral, we need to bound the covering number of the class $\mathcal{F}$ with respect to $L_2(P)$. The following lemma provides an upper bound on the covering number.

**Lemma C.4.** *There exists a universal constant $K$ such that for any VC class $\mathcal{F}$ of functions, any $r \geq 1$ and $0 < \epsilon < 1$,*

$$\sup_Q N\left(\epsilon\|F\|_{Q,r}, \mathcal{F}, L_r(Q)\right) \leq KV(\mathcal{F})(16e)^{V(\mathcal{F})}\left(\frac{1}{\epsilon}\right)^{r(V(\mathcal{F})-1)}.$$

These two lemmas are based on empirical process theory and can be found in Van der Vaart (2000).

*Proof of Theorem C.2.* By Lemmas C.3 and C.4, we have:

$$\frac{\mathbb{E}_P^*\|G_n\|_{\mathcal{F}}}{\sqrt{n}} \lesssim \sqrt{\frac{d_{VC}}{n}}.$$

Given the assumption that $\sum_{G_i} r_i(\hat{\theta}) - \sum_{G_i} r_i(\theta^*) \leq 0$, we have:

$$\mathbb{E}_{\text{in}}\left(r(\hat{\theta})\right) - \mathbb{E}_{\text{in}}\left(r(\theta^*)\right) = n^{-1}\sum_{G_i} r_i(\hat{\theta}) - n^{-1}\sum_{G_i} r_i(\theta^*)$$
$$+ \mathbb{E}_{\text{in}}\left(r(\hat{\theta})\right) - n^{-1}\sum_{G_i} r_i(\hat{\theta})$$
$$+ n^{-1}\sum_{G_i} r_i(\theta^*) - \mathbb{E}_{\text{in}}\left(r(\theta^*)\right)$$
$$\leq 2\frac{\mathbb{E}_P^*\|G_n\|_{\mathcal{F}}}{\sqrt{n}} \lesssim \sqrt{\frac{d_{VC}}{n}}.$$

Thus, we have the desired result. □

## D  Implement Details

All experiments are conducted on an NVIDIA RTX4090 GPU with 24GB memory. We implement our model based on PyTorch and PyTorch Geometric. We utilize a random splitting strategy to ensure diversity. Specifically, following previous research (Liu et al., 2023; He et al., 2024), 90% of ID samples are used for training, and 10% of ID samples and the same numbers of OOD samples are integrated for testing. For training, we utilized the Adam optimizer with an initial learning rate of $1e^{-3}$. The batch size is set to 128, and the maximum epoch is set to 400. More concretely, the hyperparameter settings for each dataset are listed in Table 4.

Table 4: Hyperparameters settings on 10 dataset pairs.

|  | optimizer | learning rate | batch size | $\beta$ | $\gamma$ | $k_1$ | $k_2$ | $k$ | $C$ |
|---|---|---|---|---|---|---|---|---|---|
| BZR+COX2 | Adam | $1e^{-3}$ | 128 | 0.3 | 1.0 | 10 | 5 | 3 | 5 |
| PTC-MR+MUTAG | Adam | $1e^{-3}$ | 128 | 0.3 | 1.0 | 10 | 5 | 10 | 3 |
| AIDS+DHFR | Adam | $1e^{-3}$ | 128 | 0.3 | 1.0 | 10 | 3 | 10 | 5 |
| ENZYMES+PROTEIN | Adam | $1e^{-3}$ | 128 | 0.3 | 1.0 | 10 | 3 | 10 | 5 |
| IMDB-M+IMDB-B | Adam | $1e^{-3}$ | 128 | 0.3 | 1.0 | 15 | 5 | 15 | 5 |
| Tox21+SIDER | Adam | $1e^{-3}$ | 128 | 0.3 | 1.0 | 10 | 3 | 15 | 5 |
| Freesolv+ToxCast | Adam | $1e^{-3}$ | 128 | 0.3 | 1.0 | 10 | 3 | 10 | 5 |
| BBBP+BACE | Adam | $1e^{-3}$ | 128 | 0.3 | 1.0 | 5 | 3 | 10 | 3 |
| ClinTox+LIPO | Adam | $1e^{-3}$ | 128 | 0.3 | 1.0 | 15 | 5 | 10 | 5 |
| Esol+MUV | Adam | $1e^{-3}$ | 128 | 0.3 | 1.0 | 10 | 3 | 10 | 5 |

## E  Ablation Study on Consistency Loss

To analyze the effect of consistency loss, we carry out experiments that compare $\mathcal{L}_{ins}$ against other commonly used contrastive losses (*i.e.,* InfoNCE loss and mean squared error (MSE) loss), and consistency loss (*i.e.,* FixMatch loss). From the results in Figure 7, we have the following observations: 1) InfoNCE and MSE generally outperform Fix-Match, as FixMatch relies on pseudo-labels derived from unreliable prediction probabilities. 2) Our Consistency loss achieves better performance than both InfoNCE and MSE,

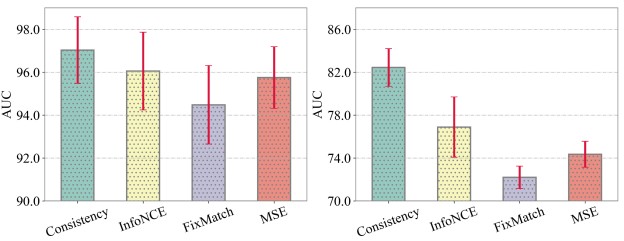

Figure 7: Ablation study on different consistency loss. We conduct experiments on two dataset pairs: BZR+COX2 (left) and IMDB-M+IMDB-B (right).

demonstrating that aligning the similarity distribution between the original graph and the augmented

## F    ADDITIONAL ANALYSIS ON PARAMETER SENSITIVITY

In this section, we conduct the parameter analysis to investigate the impact of the loss weight, i.e., $\beta$ and $\gamma$. Specifically, we search the optimal value of $\beta$ in the range of $\{0.01, 0.05, 0.1, 0.3, 0.5\}$ and $\gamma$ in the range of $\{0.1, 0.3, 0.5, 1.0, 2.0\}$. From the results shown in Figure 8, we can find that: 1) Our method is not sensitive to $\beta$. 2) The model performs poorly with a small $\gamma$, and the performance grows and begin to plateau as $\gamma$ increases, highlighting the domi-

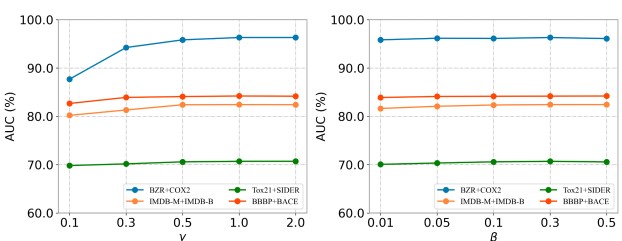

Figure 8: Sensitivity study on loss weight $\gamma$ (left) and $\beta$ (right).

nant role of our proposed $\mathcal{L}_r$, which contributes to the decision boundary refinement. These findings are consistent with the observation in ablation study. In practice, we fix $\beta = 0.3$ and $\gamma = 1.0$.

## G    COMPLEXITY ANALYSIS

The computational consumption of our model is mainly composed of three parts: (i) contextually affinitive neighborhood retrieval; (ii) consistency regularization; (iii) decision boundary compression. Given the graph with average number of node and edge $|\mathcal{V}|$ and $|\mathcal{E}|$, the number of encoder layer is $L$, the representation dimension is $d'$, and the batch size is $|\mathcal{B}|$. For (i), the time complexity of the GNN-based encoder is $O(|\mathcal{E}|Ld')$, the time complexity of the construction of global contextual $k$-NN graph is $O((|\mathcal{B}|^2 \log k_1 + k_1^2 \log k_2)d')$, the time complexity of the $k$ nearest affinitive neighborhood retrieval is $O(|\mathcal{B}|d'(k_2L + |\mathcal{B}| \log k))$. For (ii), we perform instance-aware consistency regularization from two views, which takes $O(2kd')$ for each graph. Meanwhile, given $C$ clusters in the task, the time complexity of contextual-aware consistency regularization for each graph is $O(Cd'(k+1))$. For (iii), the time complexity is $O(2d')$ for each graph. In total, the overall time complexity of CARE is $O((|\mathcal{E}|L + |\mathcal{B}|^2 \log k_1 + k_1^2 \log k_2 + |\mathcal{B}|(k_2L + |\mathcal{B}| \log k) + C(k+1) + 2)d')$. This is computationally comparable to other graph OOD methods.

