# OpenReview forum: "CARE: Contextual Affinity Exploration with Twin Concordance for Graph Out-Of-Distribution Detection"
_ICLR.cc/2026/Conference — Submitted to ICLR 2026_

### Official Review · Reviewer_7ZC5 · 2025-10-28

**Soundness:** 2
**Presentation:** 3
**Contribution:** 2
**Rating:** 4
**Confidence:** 4

**Summary:**

This paper focus on graph-level OOD detection and propose to leverage contextual affinity to better separate ID graphs from OOD ones. To achieve this, the paper extract high-order affinitive relationships between graphs and propose twin concordance learning to learn separable representations. Extensive experiments demonstrate the effectivenss of the proposed framework.

**Strengths:**

1. The motivational example in the introduction is clear and makes sense.
2. The paper well structured and the figures and tables are well ploted.
3. The prosposed method is compared with comprehensive datasets and baselines.

**Weaknesses:**

1. The construction of contextual affinity graph is problematic, it depends on GNN-generated graph representations, but I don't think GNN (random initialized)  can capture graph (semantic) affinity.
2. I believe enough graphs should be included in the batch to capture global affinity, the paper did not evaluate the influnce of different batch size. What's more, the datasets are small, I doubt its performance on large datasets as a batch can only include a small fraction of graphs.
3. The role of graph affinity in OOD dectection is not clear, making the core contribution of the paper limited.

**Questions:**

1. which readout function did you use?
2. I still don't get the effect of Perturbation-aware Concordance $L_{ins}$. As augmentations like node dropping will destroy semantics [1], this work is still leveraging topology for OOD detection.

[1] SGOOD: Substructure-enhanced Graph-Level Out-of-Distribution Detection, CIKM'24.

---

### Official Review · Reviewer_FYCP · 2025-10-29

**Soundness:** 2
**Presentation:** 3
**Contribution:** 2
**Rating:** 4
**Confidence:** 3

**Summary:**

The paper proposes CARE, an unsupervised graph OOD detection framework that jointly models contextual affinity among graphs through reciprocal nearest-neighbor relations, enhances high-order structural semantics using a meta-GNN, enforces consistency via a twin concordance scheme, and applies decision-boundary compression to better separate ID and OOD data. Extensive experiments over 10 graph OOD benchmarks and 14 anomaly detection datasets demonstrate clear improvements over recent strong baselines.

However, there still remain some concern about novelty and efficiency and causing my overall rating to 4.

**Strengths:**

1.  This paper is easy to follow and well-structured.

2. Strong empirical results which shows CARE outperforms most baselines on a wide range of datasets with both OOD and anomaly detection settings.

**Weaknesses:**

1. Although CARE combines components in a new way, many ideas resemble prior lines. For example, affinity graph has been widely used in existing Anomaly [1]  /OOD detection [2] methods.

[1] A Data-centric Framework to Endow Graph Neural Networks with Out-Of-Distribution Detection Ability

[2] Good-d: On unsupervised graph out-of-distribution detection.

2. CARE relies on per-batch affinity graph construction and meta-graph propagation, and I have concern about these will introduce non-trivial computational overhead scaling with batch size.

3. The author can include case studies showing semantic affinity improves detection beyond structural similarity.

**Questions:**

see Weaknesses

---

### Official Review · Reviewer_swdP · 2025-10-29

**Soundness:** 3
**Presentation:** 3
**Contribution:** 2
**Rating:** 2
**Confidence:** 5

**Summary:**

The paper proposes CARE, a framework for graph OOD detection that introduces contextual affinity to model relationships among graphs in a batch. It builds a contextual affinity graph, performs twin concordance learning (instance- and context-level consistency), and adds a decision-boundary compression loss. Experiments show modest improvements (1–2 % AUC) over prior methods such as HGOE and GOOD-D.

**Strengths:**

* Experiments cover several OOD benchmarks and include ablation studies.

* The framework is easy to follow.

* The method seems stable across datasets.

**Weaknesses:**

* The paper argues that prior OOD detection methods neglect contextual relations among graphs. This claim is poorly justified, since many subgraph-based or message-passing GNN approaches inherently model structural and semantic context. The motivation appears to overstate the novelty by framing a non-existent research gap.

* The paper claims that the contextual-aware concordance module uses a neural network–parameterized soft clustering function that is optimized jointly with the model to enforce semantic consistency among contextually related graphs. However, the released code shows that clustering is implemented via offline FAISS K-means on frozen embeddings, producing hard assignments (im2cluster) and static centroids. This is a standard pseudo-labeling step (as in DeepCluster-style pipelines), not a learnable differentiable module. Therefore, the key claim of end-to-end concordance learning is not substantiated by the implementation.

* Once the offline clustering step is recognized, the proposed approach reduces to a combination of well-known components: standard contrastive consistency between augmented views and an auxiliary cluster-based regularization. The “contextual affinity” concept largely renames batch-level kNN similarity used in existing contrastive GNN methods. The overall framework is incremental and lacks a clear theoretical or algorithmic advancement.

* Reported AUC gains (~1–2%) over prior work are within the expected variance of such benchmarks. No statistical tests or sensitivity analyses compared with other baselines are presented to confirm robustness or significance. Given the limited conceptual contribution, the empirical section does not convincingly demonstrate the necessity of the proposed components.

* The method is presented as a black-box aggregation of multiple losses, without explaining why contextual affinity or boundary compression improves OOD discrimination from a distributional or geometric standpoint. This makes it difficult to assess the mechanism or generalizability of the approach.

**Questions:**

* The released code relies on a module named HCL as the base model, yet this component is not mentioned, described, or cited anywhere in the paper. Moreover, the corresponding model.py file is missing, preventing reproducibility. Could the authors clarify the origin of this HCL model, its role in CARE, and why it was omitted from the manuscript?

* The implementation uses an offline FAISS K-means step for clustering, which contradicts the paper’s description of a “neural-network-parameterized soft clustering” optimized jointly with the main model. Why is the code different from the stated method, and which version (paper or code) reflects the actual results reported in Table 1?

* In the anomaly detection and ablation sections, the paper presents results without comparing against existing baselines (e.g., GOOD-D, HGOE) beyond the main OOD detection table. Could the authors explain why such comparisons were omitted and whether the proposed modules offer consistent advantages across tasks?

---

### Official Review · Reviewer_DQCU · 2025-10-30

**Soundness:** 3
**Presentation:** 4
**Contribution:** 3
**Rating:** 8
**Confidence:** 5

**Summary:**

This paper proposes a novel framework for graph OOD detection, termed CARE. The core idea is to leverage the contextual affinity of the samples for discriminative graph representations. Specifically, CARE first constructs a contextual affinity graph based on sample relationships within the hidden space. It then employs a meta-graph neural network to capture high-order affinity. Subsequently, a twin concordance learning is used to foster a robust ID boundary. Finally, a novel decision boundary compression strategy is proposed for better ID/OOD separation. Extensive experiments on ten real-world dataset pairs demonstrate the superior performance over SOTA baselines.

**Strengths:**

1. Novel idea: the proposed method opens a fresh contextual perspective for OOD detection. The core idea can be applied to various related tasks.
2. Technically sound: the motivation behind CARE is well supported by the theoretical analysis.
3. Comprehensive experimental evaluation: the proposed method is validated by extensive experiments, including comparison experiments on OOD and anomaly detection, an ablation study, and a sensitivity analysis.
4. Clear writing: the paper is well-organized and easy to follow.

**Weaknesses:**

1. The evaluation mainly emphasizes accuracy improvements. Additional metrics, such as runtime efficiency or memory consumption, are not extensively analyzed, which may impede a complete understanding of practical deployment constraints.
2. An explanation of the intuition behind the assumption in Theorem C.1 should be provided for a more solid discussion.
3. The paper states that one of four augmentation strategies is randomly employed. It's unclear if all augmentations are equally effective or if the choice of augmentation strategy significantly impacts the results.
4. Typo: The result of formula 8 is a similarity score, which is a scalar. However, it is defined as a distribution in the context.

**Questions:**

1. What is the computational cost of the proposed method? Comparison experiments in terms of running time against the most recent approaches should be provided to clearly show the trade-off.
2. The model's performance is likely sensitive to the choice and degree of data perturbation. The authors should include a sensitivity analysis on the perturbation rate to demonstrate robustness.
3. Could the authors provide an explanation of the intuition behind the assumption in Theorem C.1?
4. The CARE framework is composed of three modules: contextual affinity graph construction, twin concordance learning, and decision boundary compression. Which of these modules is the most critical for the model's overall performance gains?
5. Did the authors experiment with the impact of different augmentation strategies (e.g., using only Node Dropping vs. only Subgraph)? Is the performance robust to this choice, or is the random combination critical?

---

### Meta-Review · Area_Chair_YSDG · 2026-01-06

**Summary:**

The paper proposes CARE, a framework for graph OOD detection that leverages contextual affinity and twin concordance learning. While the idea is interesting and the paper is well-written, most reviewers raised concerns about the novelty and practical significance. Three reviewers gave negative or borderline ratings, citing issues such as incremental contribution, lack of theoretical depth, and unclear justification for key claims. Only one reviewer recommended acceptance, mainly due to strong empirical results.

**Reviewer Concerns:**

The authors did not submit a rebuttal, so none of the concerns were addressed. Key issues include:

+ Novelty: Several components resemble prior work, and the claimed research gap is not well justified.
+ Implementation mismatch: The code uses offline clustering, contradicting the paper’s claim of end-to-end learning.
+ Lack of efficiency analysis and sensitivity studies.
+ Limited explanation of why contextual affinity improves OOD detection beyond existing methods.

**Reviewer Scores:**

Given the absence of rebuttal and discussion, I think the reviewers would maintain their original scores:
- Reviewer DQCU (positive): Likely unchanged (8).
- Reviewer swdP (strong reject): Unchanged (2).
- Reviewer FYCP (borderline reject): Unchanged (4).
- Reviewer 7ZC5 (borderline reject): Unchanged (4).

---

### Decision · Program_Chairs · 2026-01-26

Reject